# Mesophilic and Thermophilic Anaerobic Digestion of Organic Fraction Separated during Mechanical Heat Treatment of Municipal Waste

**Slawomir Kasinski**

Department of Environmental Biotechnology, University of Warmia and Mazury in Olsztyn, Sloneczna St. 45G, 10-709 Olsztyn, Poland; slawek@kasinski.pl; Tel.: +48-604-580-234

**Abstract:** The objective of this study was to investigate the effect of process temperature on semi-continuous anaerobic digestion of the organic fraction separated during autoclaving of municipal waste. Tests were carried out in reactors with full mixing. Biogas production was higher in thermophilic conditions than in mesophilic conditions (0.92 L/g volatile solids at 55 °C vs. 0.42 L/g volatile solids at 37 °C, respectively). The resulting methane yields were 0.25-0.32 L $CH_4$/g VS and 0.56–0.70 L $CH_4$/g VS in mesophilic and thermophilic conditions, respectively. In both variants, the methane share was over 70% *v/v*. This work also discusses the potential impact of Maillard compounds on the efficiency of the fermentation process, which were probably produced during the process of autoclaving of municipal waste. These results indicate that, after autoclaving, the organic fraction of municipal waste can be an effective substrate for anaerobic digestion in thermophilic conditions.

**Keywords:** Mechanical Heat Treatment; autoclaving; autoclaving of municipal waste; mesophilic fermentation; thermophilic fermentation; anaerobic digestion

## 1. Introduction

Mechanical Heat Treatment (MHT) is a relatively new and poorly investigated technology for processing municipal solid waste. MHT uses a combination of mechanical and thermal-based technologies to separate a waste stream into several component parts and enable further options for recycling and biological treatment. MHT sterilizes pathogens, deodorizes the waste stream, reduces waste mass (mainly by dehydration), compacts plastics and disintegrates labels on glass bottles, food packaging and cans. The separated utility fractions produced by MHT, including pre-residual derived fuel (pre-RDF), account for up to 80% of the initial waste mass [1].

The most common method of thermally treating municipal waste is autoclaving. This technology is used by the majority of existing European MHT plants operating on a technical scale [2]. Technologically, the MHT process consists of a grinding stage (I), autoclaving (II) and mechanical separation of material fractions (III). The largest fraction obtained during MHT is the organic remaining fraction (ORF), mechanically separated after full-scale autoclaving of unsorted municipal solid waste (Figure 1). Depending on the technological process and the composition of the waste, this fraction constitutes up to 61.5% of the initial weight of waste [3]. This fraction consists mainly of thermally processed organic waste, such as paper, kitchen and garden waste and other organics, which are turned into a very homogenous fibrous material under the influence of temperature [4]. According to Papadimitriou [1], a properly conducted MHT process enables approximately 80% of the initial waste stream to be recycled/recovered. However, this level of effectiveness can only be obtained if the entire organic fraction remaining after the process of autoclaving is managed.

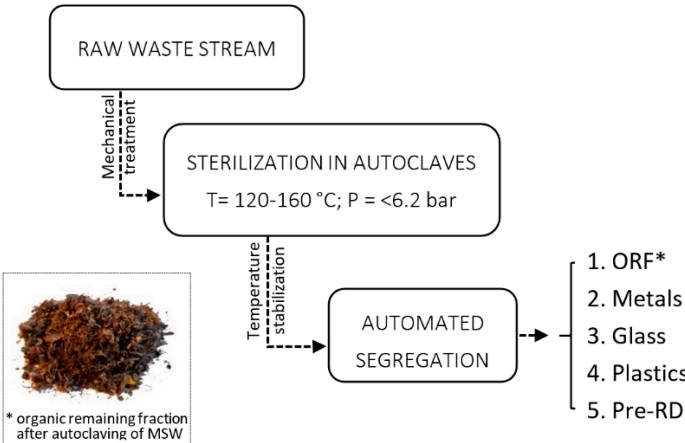

**Figure 1.** Flow-chart illustrating a process for Mechanical Heat Treatment of Municipal Solid Waste.

Although autoclaving dissolves paints on cans and other packages and thus could contribute to heavy metal pollution of organic fiber [4,5], contamination resulting from autoclaving is similar to that from the most effective mechanical–biological treatment systems [4]. The process of autoclaving disrupts the lignocellulosic structure in organic waste [6], which theoretically facilitates the decomposition of organic matter, both in aerobic and anaerobic conditions. A test performed by Wojnowska-Baryla et al. [7] showed that the effect of waste sterilization during autoclaving is not permanent, and the organic remaining fraction converts to biologically unstable material that could not be landfilled without posttreatment (ca. AT4 = 25 g $O_2$ kg$^{-1}$ TS). The separated ORF can be also contaminated with glass and other foreign matter, which limits the production of high-quality compost [8]. For this reason, research on the aerobic treatment of the autoclaved organic fraction is being conducted with the goal of achieving aerobic stabilization, i.e., in passive aeration conditions [7]. Another test, performed by Kulikowska et al. [9], showed that stabilizate from autoclaved municipal solid waste might be a source of valuable humic substances (HS). The maximum HS production of 82–120 mg/g OM was similar to that in composts from different kinds of organic waste, which partly justifies the practical use of the autoclave process in a waste circular economy.

Due to the above-mentioned problems with compost management after autoclaving, there is interest in anaerobic digestion of autoclaved organic wastes. However, research done so far has usually been carried out in batch conditions, with the use of agricultural waste or synthetically prepared organic waste. Menardo et al. [10] investigated the impact of the autoclaving process on anaerobic digestion of swine slurry and obtained 115% higher production of methane when the slurry was autoclaved at 120 °C. Similarly, when waste activated sludge was digested in a CSTR reactor, methane production was 12% and 25% higher after autoclaving at 135 °C and 190 °C, respectively, than without autoclaving [11]. In contrast, some data indicates decreased biogas production and biodegradability of autoclaved organic mass. Tampio et al. [12] compared the efficiency of anaerobic digestion of autoclaved (160 °C, 6.2 bar) and untreated food waste in semi-continuously fed mesophilic reactors. The authors observed lower specific methane yield during digestion of autoclaved substrate even after a long period of microbial acclimatization (473 days of the process). Similar results were obtained by García et al. [5], who compared the efficiency of the process of anaerobic digestion of autoclaved (145 °C, 600 kPa) and untreated source-separated OFMSW (organic fraction of municipal solid waste). Both Tampio et al. [12], as well as García et al. [5], indicate that reduced biogas production can be caused by Maillard compounds produced during the autoclaving process.

It should be clearly stated that there is a lack of research using autoclaved organic municipal waste, especially that obtained from installations operating under technical conditions. Due to the fact that MHT is a technology in the developmental phase, there is a need to expand knowledge in this field. The uniqueness of the organic remaining fraction obtained during MHT of municipal waste, including

its homogeneity and fibrous structure, makes it impossible to determine the impact of the autoclaving process on anaerobic digestion of ORF. However, anaerobic digestion of ORF obtained during the autoclaving process still is an interesting and noteworthy topic. Therefore, the objective of this study was to investigate the effect of process temperature on anaerobic digestion of the organic remaining fraction mechanically separated after full-scale autoclaving of unsorted municipal solid waste. The tests were carried out in mesophilic and thermophilic conditions in reactors with full mixing.

## 2. Materials and Methods

### 2.1. Methods of Sampling and Characteristics of Test Material

Substrates for anaerobic digestion were obtained from the organic fraction of municipal waste separated in the MHT plant in Różanki (Poland). Autoclaving was preceded by the separation of large waste items from the mixed waste stream and mechanical homogenization (pre-preparation) to achieve grain sizes of <350 mm. The pre-prepared waste was treated in autoclaves under saturated steam conditions, at 120 °C–150 °C and 2–5 bar. The production and condensation of steam took place in a closed system. Mechanical treatment took place after the autoclaving process to separate recyclable materials. After autoclaving, the separated ORF had a particle size of <10 mm and comprised ca. 30% of the waste.

A sample of ORF for anaerobic digestion was collected from a randomly formed heap of about 5 m$^3$, from which the laboratory sample was separated by the trapezoidal diverging method [13]. In accordance with the methodology, the sample was divided into three laboratory parts with a total volume of 45 L, which were then transported in sterile containers to the testing site. One part was subjected to basic physico-chemical and respirometric tests. The remaining two parts were ground to a grain size of <1 mm and then used as the research substrate. The grinding was necessary to facilitate feeding of the waste into the reactor through the narrow hole of the shutoff valve. Additionally, according to Izumi et al. [14], grinding enhances the solubilization of organic matter, which was important during the preparation of the feeding mixture (described in *Section 2.3*). Grinding also affects the process of methanization of organic matter [14].

The physico-chemical properties of ORF are presented in Table 1. The ORF sample for anaerobic digestion had a neutral pH and an organic substance content of 58% DM. Morphological analysis indicated that this fraction was homogenous and had a glass share of 1–2% by mass, which limits its use as soil conditioner. The fraction showed high hygroscopicity—opening a container with a sample caused the dry matter content to decrease from 84% to 69% within 7 days. At 31% humidity, the AT4 value was 26.09 g $O_2$/kg DM, which indicated the need to stabilize the fraction prior to storage. The analytical methods used for physico-chemical characterization of ORF are described below in Section 2.4.

**Table 1.** Characteristics of organic remaining fraction (ORF) after autoclaving, fraction <10 mm.

| Characteristic | Unit | Value |
|---|---|---|
| pH | - | 7.5 |
| COD (soluble) | mg/L | 4140 |
| Total Nitrogen (N) | *% w/w* | 1.07 |
| Ammonia Nitrogen | *% w/w* | 0.08 |
| Phosphorus | *% w/w* | 0.50 |
| Total solids | *% w/w* | 84 |
| Volatile solids | % DM | 58 |
| AT4 | mg $O_2$/g DM | 26.09 |

### 2.2. Test Stand

Anaerobic digestion experiment was carried out in accordance with VDI [15] and DIN [16] standards. The tests were carried out in two stainless steel fermenters with full mixing and an active

volume of 6 L. The fermenters were equipped with an agitator with adjustable rotation speed and a heating mantle. Properly located valves enabled substrate feeding and collection of post-fermentation residues. The produced gas was collected with a separate valve in Tedlar bags. The reactors worked in a semi-continuous system and were fed with the sterile waste substrate once a day after the collection of post-fermentation waste. A scheme of the reactors is presented in Figure 2.

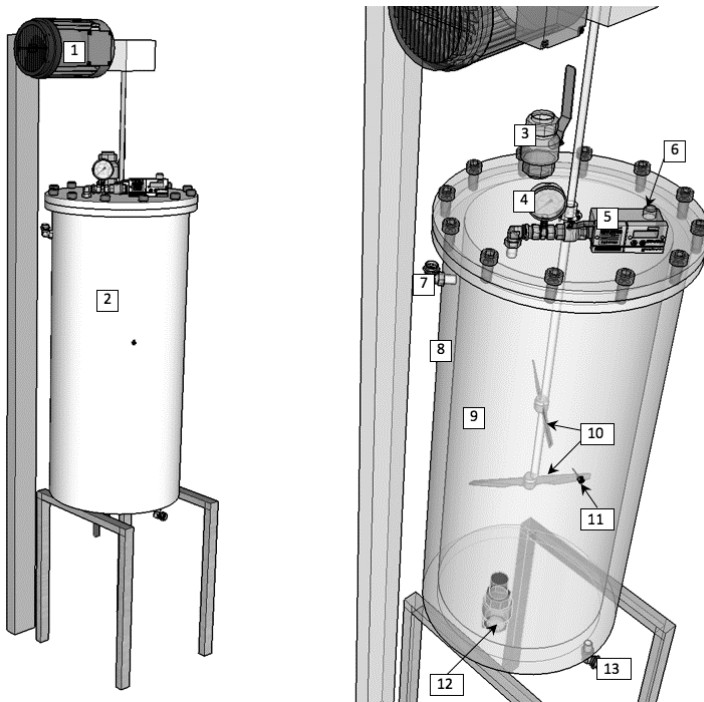

**Figure 2.** Experimental stand. (1) Electric engine, (2) fermenter with tooling, (3) shutoff valve/substrate feeder, (4) manometer, (5) biogas flow meter, (6) biogas outlet, (7) water jacket feed valve, (8) watercoat, (9) reactor chamber, (10) stirrers, (11) PT-100 temperature sensor, (12) shutoff valve/post-fermentation outlet, (13) water jacket drain valve.

### 2.3. Digestion Set-up

Technological studies were carried out for 90 days in two technological variants differing in process temperature. Variant 1 was conducted at 37 °C; variant 2, at 55 °C. In both variants, the organic loading rate (OLR) and the hydraulic retention time (HRT) were 2 g VS/L·d and 30 d, respectively.

The feed was the ground ORF separated by MHT of municipal waste and mixed with tap water at a ratio of 1:7. The reactor was inoculated with anaerobic sludge originating from the municipal wastewater treatment plant in Olsztyn (north of Poland, population 175,000). The characteristics of the inoculum were as follows: pH, 7.31; COD, 1520 mg/L; ammonium, 88 mg NNH4/L; alkalinity, 77 mval/L; concentration of total solids, 76.2 g/L; phosphate, 409 mg P-PO4/L. For acclimatization to the temperature in the reactor, the inoculum was left in the reactor for 20 days, after which, dosing of the substrate to the reactor began.

### 2.4. Analytical Monitoring of the Process

The separated ORF after autoclaving was analyzed in terms of pH, COD, total nitrogen, ammonium nitrogen, phosphorus, total solids and volatile solids. The analyses were performed by the Institute of Soil Science and Plant Cultivation (IUNG), State Research Institute in Puławy, Poland. As presented above, at the start of the experiment, the inoculum was analyzed to measure the pH, COD, ammonium, alkalinity, total solids concentration and phosphates.

The post-fermentation residues were analyzed in terms of total solids and volatile solids concentration. The liquid phase of the post-fermentation residues (obtained by centrifuging the digestate samples for 10 min at 9000 rpm) was analyzed with regard to pH, COD, ammonium nitrogen, phosphates and volatile fatty acids (VFAs). The analyses of post-fermentation residues were performed at one-day intervals during the first 20 days and then at two-day intervals.

Analyses of total solids, volatile solids, COD, alkalinity, ammonium nitrogen and phosphates conducted by IUNG and in our laboratories were all performed according to standard methods for examination of water and wastewater, APHA [17]. According to the methodology of the IUNG, total nitrogen was analyzed by elemental analysis using a FLASH 2000 elemental analyzer (Thermo Scientific, USA), phosphorus content was analyzed by the spectrophotometric method based on an ammonium molybdate reaction.

VFAs (acetate, propionate, isobutyrate, butyrate, isovalerate, valerate, isocaproate and caproate) were analyzed using a gas chromatograph (GC, Varian 3800, Australia) equipped with a capillary column (Factor-Four VF-1 ms, 30 m × 0.25 mm i.d., 1.0 lm film; Varian). A volatile acid standard mix (Supelco, USA) was used as the standard. The flame ionization detector (FID) was operated at 280 °C with an injection port temperature of 250 °C. The split ratio was 1:100. The initial column temperature was set to 100 °C, then raised at a rate of 20 °C/min to 200 °C and finally held at 200 °C for 1 min. Helium was used as the carrier gas at a flow rate of 1.0 mL/min. Samples were prepared for VFA analysis according to Gilroyed et al. (2010).

To determine the biological stability of the ORF, the AT4 analysis was performed. The AT4 analysis (performed in triplicate) is a respiration test employing a manometric method, which was performed with the Oxitop Control measuring system. Waste samples were incubated at 20 °C in the presence of a 1M sodium hydroxide solution (for $CO_2$ absorption) for four days.

The volume and composition of the biogas produced during fermentation were monitored. This analysis was carried out daily on averaged samples collected in Tedlar bags. The gas composition, including oxygen, carbon dioxide and methane content, was measured using a GA 2000 PLUS gas analyzer (Geotech).

## 3. Results and Discussion

During the first 50 days of fermentation, the biogas productivity was similar in both mesophilic (37 °C) and thermophilic conditions (55 °C): 3154–7803 mL/d and 3365–6874 mL/d, respectively, corresponding to yields of 0.26–0.65 L/g VS and 0.28–0.57 L/g VS, respectively. After 50 days, biogas productivity was higher in thermophilic conditions than in mesophilic conditions. In thermophilic conditions, the productivity first increased to about 12,000 mL/d (2000 mL/L), then stabilized at about 11,000 mL/d (0.92 L/g VS). In mesophilic conditions, the biogas productivity was about 5000 L/d (0.42 L/g VS). After initial fluctuations, in both variants, the production of biogas stabilized on about the 70th day of the process. The resulting methane yields were 0.25–0.32 L $CH_4$/g VS and 0.56–0.70 L $CH_4$/g VS in mesophilic and thermophilic conditions, respectively (Figure 3).

This difference in biogas productivity between mesophilic and thermophilic conditions is consistent with what has been reported in the literature [18]. Interestingly, however, the biogas productivity in the present study is higher than that in other studies. For example, Cavinato et al. [19] reported that, during co-fermentation of waste activated sludge with biowaste, the specific biogas production increased from 0.34 to 0.49 L/g TVS when the reactor temperature was changed from mesophilic (37 °C) to thermophilic (55 °C). The final values of methane yield, however, seem to be higher than those reported in the literature. Forster-Carneiro et al. [20] examined anaerobic digestion under thermophilic conditions (55 °C) of three source-separated organic fractions of waste: food waste, OFMSW and shredded OFMSW. The authors obtained methane yields of 0.18 L $CH_4$/g VS, 0.05 L $CH_4$/g VS and 0.08 L $CH_4$/g VS, respectively. Zhang et al. [21], during batch anaerobic digestion of source-separated food wastes, obtained 0.43 L $CH_4$/g VS. Interestingly, the authors used grinding and freezing as a

method of pre-treatment and noted that the average methane content of biogas was 73%, similar to the present results.

In the initial phase of fermentation, the methane content in the biogas increased to around 70% and 60% *v/v* at 37 °C and 55 °C, respectively. Then, regardless of the process temperature, methane synthesis slowed. At 37 °C, the slowdown took place between the 12th and 44th day; and at 55 °C, between the 12th and 50th day. In the stabilization phase, the methane content was about 75% *v/v* in both variants (Figure 4).

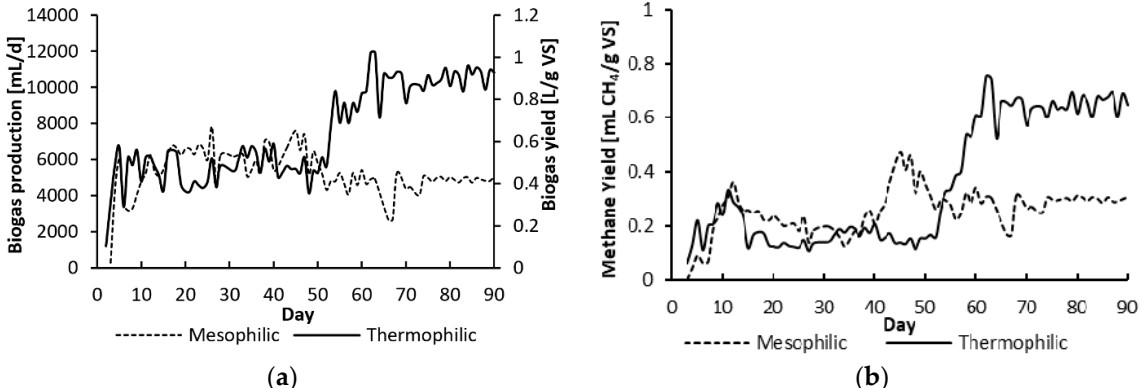

**Figure 3.** Biogas production (**a**) and methane yield properties (**b**) obtained during the anaerobic digestion of ORF.

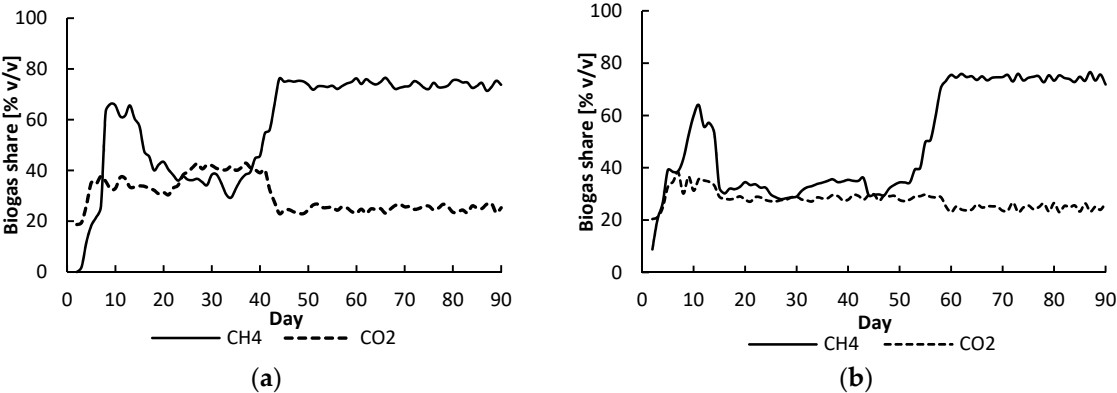

**Figure 4.** Changes in the biogas composition during the anaerobic digestion of ORF; (**a**) mesophilic—temperature 37 °C, (**b**) thermophilic—temperature 55 °C.

In the present study, the short burst of intense methanogenic activity at the start of the process can be explained by the fact that anaerobic digester sludge was used as the inoculum. This type of sludge has been reported to be the best inoculum for a rapid start-up, as it has much higher numbers of aceticlastic methanogens than other start-up materials, such as cattle manure [22]. The subsequent slowdown in methane synthesis in both research variants may be related to the need for the microorganisms to adapt to the new substrate, as the methane share did not correlate with the other process parameters that were monitored. It is interesting to note, however, that not only was the proportion of methane high in comparison to other literature reports [23,24], but the methane content was high in both variants despite substantial differences in the process parameters (pH, VFA concentration).

The VFA concentrations differed in the two variants. At 37 °C, the VFA concentration first increased, then decreased and stabilized at about 600 mg VFA/L. At 55 °C, the VFA concentration increased from day 20 of the process, after which it ranged from 1500 to 2500 mg/L until it stabilized at 2400–2600 mg/L. In both reactors, the concentration of propionic acid was higher than that of acetic acid. From day 20 of the process, the acetic acid concentration was around 5.5-times higher at 55 °C

than at 37 °C. At 55 °C, the concentrations of caproic and isobutyric acids were 147–184 mg/L and 84–101 mg/L, respectively. At 37 °C, in contrast, these acids were present only during the earlier stages of the process and their concentrations were relatively low (Figure 5).

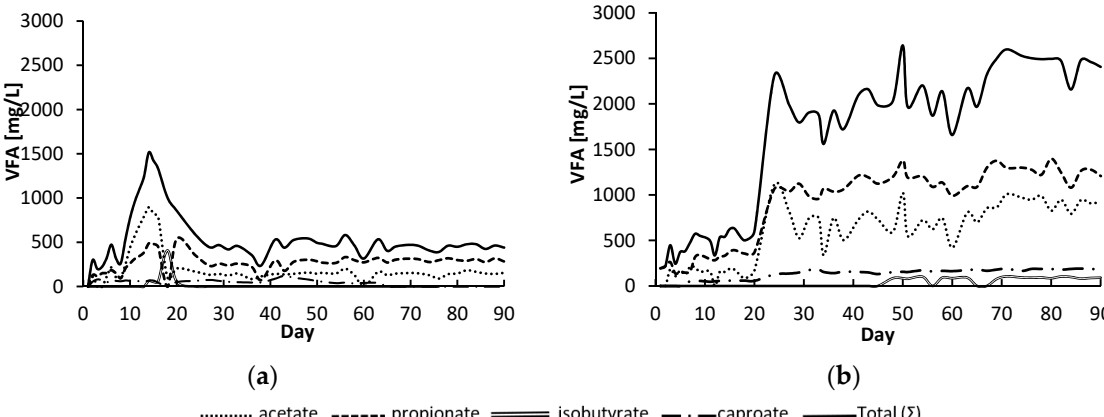

**Figure 5.** Qualitative and quantitative changes of volatile fatty acids (VFA) during the anaerobic digestion of ORF after autoclaving; (**a**) mesophilic—temperature 37 °C, (**b**) thermophilic—temperature 55 °C.

At both operational temperatures, propionate constituted the majority of the VFAs. Similar results were obtained by Kim et al. [25] during co-digestion of anaerobic sludge and dog food in a CSTR reactor and by Labatut et al. [26] during co-digestion of cow manure and dog food in a CSADs reactor. Labatut et al. [26] attributed propionate accumulation to free energy limitations imposed by molecular hydrogen. According to the authors, thermophilic bacteria that oxidize LCFA have a higher PH2 threshold than their mesophilic counterparts (10–2.4 atm vs. 10–3 atm, respectively). Thus, LCFAs can be degraded to a greater extent at thermophilic temperatures, potentially producing additional $H_2$ and further inhibiting propionate oxidation. This, compounded by faster hydrolysis rates, can make thermophilic digesters more susceptible to propionate accumulation, and consequently process upsets. However, this is only speculation in the case of the present study, as the partial hydrogen pressure was not measured during the tests.

Another interesting hypothesis concerns the effect of compounds formed during Maillard's reaction on the efficiency of methanogenesis. These compounds are produced by reactions between amino acids and carbohydrates at temperatures above 100 °C [27,28], and they inhibit microbial activity [29,30]. It can, therefore, be assumed that such compounds were formed during autoclaving of the municipal waste used in the present study. This may well have lowered the efficiency of anaerobic digestion in mesophilic conditions, as reported by Tampio et al. [12] and García et al. [5]. In addition, the efficiency of anaerobic digestion under mesophilic conditions was not reduced in the present study, as also reported by Nakajima-Kambe et al. [31]. This suggests that, under thermophilic conditions, the microorganisms adapted to the presence of Maillard compounds. Considering that the methanogens' activity was not lowered, an explanation would be the inhibitory effect of Maillard compounds on the acetogenesis process at 37 °C, which would result in less biogas production with a higher methane content. However, it should be remembered that other inhibitors, such as furfural, hydroxymethylfurfural (HMF) or phenols could also be formed during the autoclaving process. Thus, further investigations would be necessary to indicate which inhibitors were present.

During fermentation at 37 °C, the pH remained between 7.84 and 8.33. At 55 °C, it was higher and ranged from 8.03 to 8.51. Similar results were obtained by Song et al. [32] during single-stage mesophilic and thermophilic digestion of sewage sludge. Normally, increased gas production accompanies increasing pH because methanogenesis reduces VFA and produces alkalinity. In the presented studies, the changes in VFA did not affect the pH changes during fermentation. Song et al. [32] indicate that the increase in pH was caused by the degradation of nitrogenous compounds; nevertheless,

in the present studies, such an association was not found (Figure 6). In both research variants, an increase in the concentration of dissolved organic compounds in the reactors was observed. At 37 °C, the increase occurred in the first 15 days and was on average 218 mg COD/L·d. After this time, the COD concentration stabilized at a level between 3200 and 3680 mg COD/L. At 55 °C, the concentration of organic compounds dissolved on day 70 of the process increased by an average of 52 mg/L·d. After this time, the concentration stabilized within the range of 4840–5000 mg/L. From day 70 of the process, COD concentration was about 1.4 times higher at 55 °C than at 37 °C (Figure 6). The course of changes in the concentration of ammonium nitrogen in both reactors was similar. After an initial increase, the ammonium nitrogen concentration dropped, then increased starting on day 30 and, finally, stabilized from the 70th day at 169–207 mg/L and 262–304 mg/L at 37 °C and 55 °C, respectively. The initial concentration of orthophosphate was two-fold lower at 37 °C than at 55 °C, and it decreased at a rate two-times slower (0.63 mg/L·d). The final concentrations of orthophosphate were 19.96 mg/L and 28.43 mg/L at 37 °C and 55 °C, respectively (Figure 6). The initial increase in nitrogen concentration was related to the rapid biodegradation of the organic matter contained in the autoclaved municipal waste. The slowdown of the biodegradation process from the 30th day of the process may have been due to the inhibitory effect of ammonium nitrogen on the fermentation process [33,34], the effect of Maillard compounds, or the effect of other inhibitory compounds formed during the autoclaving process. From the 70th day of the process, ammonium nitrogen and other process indicators (COD, phosphate, VS) stabilized, which confirms that the microorganisms had adapted to the process conditions, which is similar to the findings of Hashimoto [35].

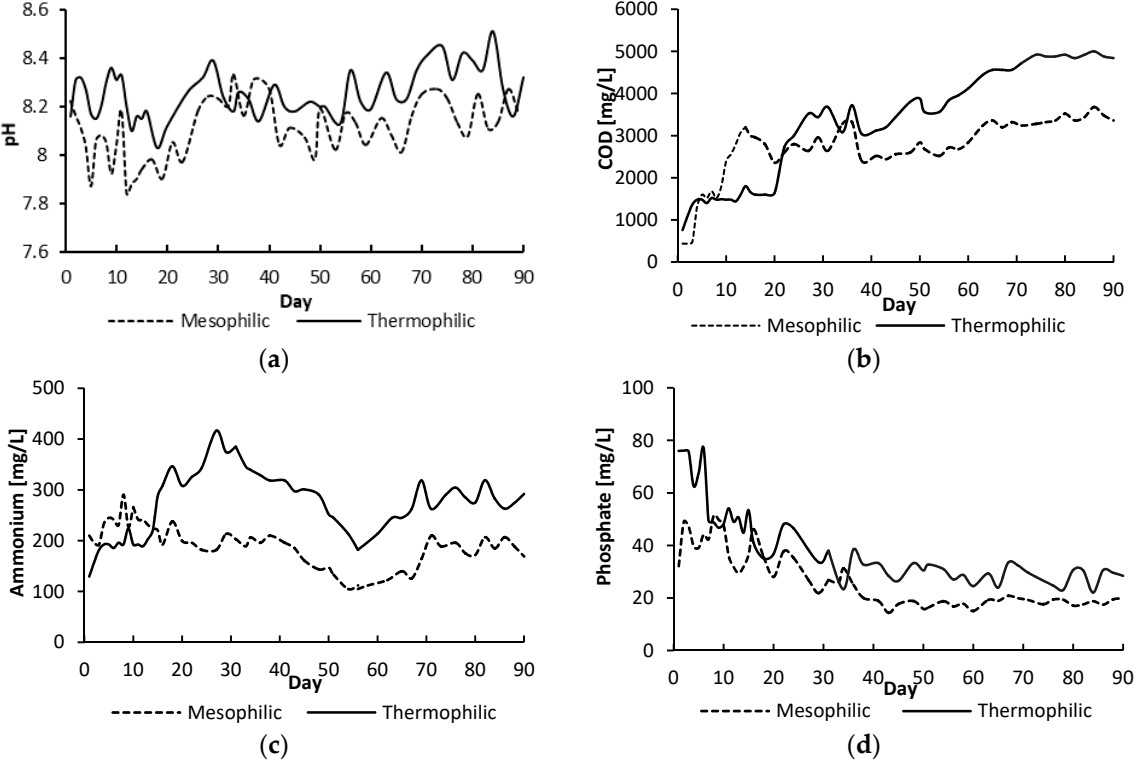

**Figure 6.** Changes of pH (**a**), dissolved organic matter (COD) (**b**), ammonium nitrogen (**c**) and orthophosphates (**d**) during the anaerobic digestion of ORF after autoclaving.

The results of Fisher's exact test ($p = 0.29609$) did not show a significant effect of temperature on the efficiency of organic compounds removal. The efficiency of organic compounds removal in the initial stage was above 90%. After 70 days of anaerobic digestion, it dropped to 67–69% at 37 °C and 68–71% at 55 °C. Until the first complete replacement of the reactor volume (day 30), the average reduction in organic compound removal at 37 °C was 0.53%/d, after which it was 0.08%/d, although the technological conditions

remained the same. In the second variant, the changes were slower: 0.37%/d and 0.1%/d, respectively. The reduction in the use of organic compounds by microorganisms resulted in an increase in dry matter in the reactors. During the first 30 days, the changes took place more dynamically: at 37 °C, the organic dry matter content increased from 6.02 to 12.77 g/L; and at 55 °C, from 3.67 to 12.54 g/L. From the 70th day of the fermentation process, stabilization of the organic dry matter content was observed in both reactors at 18.39–20.14 g/L and 17.67–19.34 g/L, respectively (Figure 7).

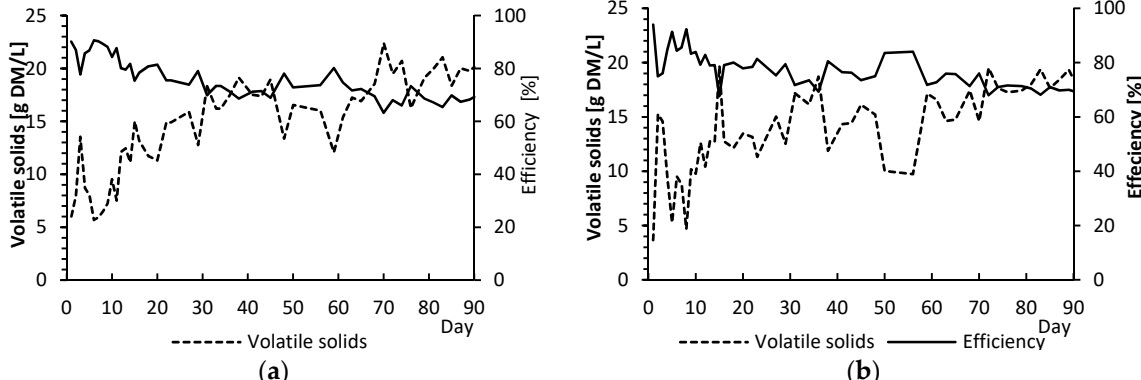

**Figure 7.** Efficiency of removal of organic compounds during the anaerobic digestion of ORF after autoclaving; (**a**) mesophilic—temperature 37 °C, (**b**) thermophilic—temperature 55 °C.

The lack of a significant influence of temperature on the intensity of the transformation of organic compounds seems to be consistent with other reports. Similar results were obtained by Song et al. [26] during a single stage of mesophilic and thermophilic fermentation of sewage sludge. In the present study, ammonium nitrogen accumulated along with an increase in VS. This is particularly evident under thermophilic conditions, in which accumulation of VS and ammonium nitrogen occurred on the 30th day of the process (correlation coefficient = 0.71). In mesophilic conditions, no correlation between these parameters was found.

## 4. Conclusions

The obtained results indicate that temperature influences the efficiency of the fermentation process. The process carried out under thermophilic conditions (55 °C) was characterized by higher biogas productivity, higher production of fatty acids, higher pH, higher concentration of chemical compounds and higher efficiency of organic compounds removal. In the present study, the biogas productivity at 55 °C at the time of process stabilization was 0.92 L/g VS, which was double the productivity at 37 °C (0.42 L/g VS). This is not surprising, considering that overall digestion rates can be up to 2.25 times faster at thermophilic temperatures than at mesophilic ones [18]. It should be assumed that the adaptation of microorganisms to the substrate took place around the 70th day of the process. Taking into account the characteristics of the substrate given in Table 1, biogas productivity was 0.20 $m^3$/Mg FM and 0.45 $m^3$/Mg FM, which is particularly interesting for application purposes on a technical scale.

This study has shown that the organic fraction of municipal waste separated during MHT of municipal waste can be an effective substrate for anaerobic digestion in thermophilic conditions.

**Funding:** Project financially supported by Minister of Science and Higher Education the range of the program entitled "Regional Initiative of Excellence" for years 2019–2020, project No. 010/RID/2018/19, amount funding 12.000.000 PLN.

**Conflicts of Interest:** The authors declare no conflicts of interest.

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
