# Peer review of "Mesophilic and Thermophilic Anaerobic Digestion of Organic Fraction Separated during Mechanical Heat Treatment of Municipal Waste"

_applsci, doi:10.3390/app10072412_

Round 1

Reviewer 1 Report

The article is very interesting. however some additional considerations are needed:

#1: in the intruduction section  the difference between the collection system of OFMSW should be added (differently sorted OFMSW). Further the heavy metals in mechanicaly separated OFMSW can cause the inhibition and the decrease the quality of digestate. Consider including:

Cecchi, F., Bolzonella, D., Pavan, P., Macé, S., Mata-Alvarez, J., 6.36, Anaerobic Digestion of the Organic Fraction of Municipal Solid Waste for Methane Production, Research and Industrial Application

Colazo, A. B., Sánchez, A., Font, X., Colón, J., Environmental impact of rejected materials generated in organic fraction of municipal solid waste
anaerobic digestion plants: Comparison of wet and dry process layout.

#2: line 69-71: methane fermentation - would prefer anaerobic digestion

3:74-75: The OFMSW should be determined regarding to sorting (see Cecchi and Colazo). Characteristics of inoculum shoud be included. How was the size of the grain determined? Sieve analysisi, particle size analysis? - this is missing and should be included in materials and methods.

4: quasi-continuous = semi-continuous. How often? 1/day, 2/day?

5: line110-112: fermented= digested, chambers= reactors, methane fermentation=anaerobic digestion. What kind of adaptation of inoculum?

6: line119-125. no methods are provided for the chemical analyses. (AHPHA?).

7: The results obtained should be compared to other research. for example: Murovec, B., Kolbl, S., Stres, B.; Methane Yield Database: Online infrastructure and bioresource for methane yield data and related metadata

De Mes, T. Z. D., Stams, A. J. M., Reith, J. H., and Zeeman, G., Methane Production by Anaerobic Digestion of Wastewater and Solid Wastes,

8: how did the particle size (grinding to the 1 mm particles) effect the pmethane production? Ginding is mechanical pretreatment, increasing surface area, decreasing volume... (see Kouici Izum et al., 2010, Kolbl et al. 20149, Hajji et al, 2013).

9: 162-165: the agricultural inoculum is also suitable for start-up as it has plethora of needed microorganisms that are essential for anaerobic digestion. (research of full scale anaerobic digestion start-up Kolbl Repinc et al., 2018).

10: line 221-222 rate of liquefaction of organic compounds? how was this determined?

11:line 241-243 and line256 : significant effect? statistical analyses should be included to provide data on sigificant differences/no significant differences

Author Response

#1: in the intruduction section  the difference between the collection system of OFMSW should be added (differently sorted OFMSW). Further the heavy metals in mechanicaly separated OFMSW can cause the inhibition and the decrease the quality of digestate.

The manuscript unintentionally misused the phrase OFMSWA, and therefore the substrate used in the study may have been understood as source-separated OFMSW. In order to solve this problem, the wording organic fraction of municipal solid waste after autoclaving (OFMSWA) has been replaced with organic remaining after autoclaving – ORF (organic remaining fraction mechanically separated after full-scale autoclaving). The changes were made throughout the whole manuscript.

Lines (Track changes, simple markup): 34, 52, 81, 95, 97, 107, 113, 116, 135, 143, 167, 187, 207, 229, 287, 300,

To emphasize the uniqueness of the obtained organic fraction, the Introduction section has been rewritten and a diagram of the process has been added.

Lines (Track changes, simple markup): 23-86, diagram: 42

#2: line 69-71: methane fermentation - would prefer anaerobic digestion

This change has been made throughout the manuscript.

Lines (Track changes, simple markup): 11, 18, 20, 61, 64, 69, 73, 81, 83, 89, 97, 107, 108 187, 194, 207, 229, 248, 229, 287, 290, 300, 322

3:74-75: The OFMSW should be determined regarding to sorting (see Cecchi and Colazo). Characteristics of inoculum shoud be included. How was the size of the grain determined? Sieve analysisi, particle size analysis? - this is missing and should be included in materials and methods.

The OFMSW misunderstanding was explained above (1 #). The characteristics of the inoculum have been added to the manuscript.

Lines (Track changes, simple markup): 137-139

The method of obtaining waste with a specific grain size has been shown in Figure 1.

Lines (Track changes, simple markup): 42

4: quasi-continuous = semi-continuous. How often? 1/day, 2/day?

The wording “quasi-continuous” has been corrected to “semi-continuous”. Information about the frequency is given at the end of the sentence (Line 123).

Lines (Track changes, simple markup): 10, 11, 69, 70, 123

5: line110-112: fermented= digested, chambers= reactors, methane fermentation=anaerobic digestion. What kind of adaptation of inoculum?

The appropriate changes have been made throughout the manuscript. The paragraph also explains how the inoculum adaptation process was performed.

Lines (Track changes, simple markup): 139-141

6: line119-125. no methods are provided for the chemical analyses. (AHPHA?).

The Analytical monitoring of the process section has been completely reorganized.

Lines (Track changes, simple markup): 143-174

7: The results obtained should be compared to other research.

An additional literature review has been added. Additional calculations of methane yield were made. The results were given, presented in the appropriate graph (Figure 3b) and compared with the literature.

Lines (Track changes, simple markup): 183-186

The following papers have been cited:

Forster-Carneiro, T.; Pérez, M.; Romero, L. I. Thermophilic anaerobic digestion of source-sorted organic fraction of municipal solid waste. Bioresource technology 2008, 99 (15), 6763-6770.

Zhang, R.; El-Mashad, H. M.; Hartman, K.; Wang, F.; Liu, G.; Choate, C.; Gamble, P. Characterization of food waste as feedstock for anaerobic digestion. Bioresource technology 2007, 98 (4), 929-935.

Lines (Track changes, simple markup): 194-200

The selection of these papers was also based on the comments of the other reviewers.

8: how did the particle size (grinding to the 1 mm particles) effect the pmethane production? Ginding is mechanical pretreatment, increasing surface area, decreasing volume... (see Kouici Izum et al., 2010, Kolbl et al. 20149, Hajji et al, 2013).

I am particularly grateful for the above comment. The grinding of waste was of a technical nature. The grinding was necessary to facilitate feeding of the waste into the reactor through the narrow hole of the shutoff valve. An appropriate explanation was made and, additionally, the effect of grain size on the fermentation process was described, according the above comment. The following paper was cited:

Izumi, K.; Okishio, Y.K.; Nagao, N.; Niwa, C.; Yamamoto, S.; Toda, T. Effects of particle size on anaerobic digestion of food waste. International biodeterioration & biodegradation 2010, 64 (7), pp. 601-608.

Lines (Track changes, simple markup): 102-106

9: 162-165: the agricultural inoculum is also suitable for start-up as it has plethora of needed microorganisms that are essential for anaerobic digestion. (research of full scale anaerobic digestion start-up Kolbl Repinc et al., 2018).

I fully agree; however, the aim of this work was not to analyze the impact of different kinds of inoculum on the process of anaerobic digestion.

10: line 221-222 rate of liquefaction of organic compounds? how was this determined?

This wording was incorrect and has been removed.

Lines (Track changes, simple markup): 268-269

11:line 241-243 and line256 : significant effect? statistical analyses should be included to provide data on sigificant differences/no significant differences

An appropriate statistical analysis was carried out (Fisher's exact test)

Lines (Track changes, simple markup): 288

Reviewer 2 Report

This manuscript reports the anaerobic digestion of the organic fraction of municipal waste under meso- and thermophilic conditions. This work deals with a very specific and narrow topic, which might be of some interest if conducted properly. There are many major concerns before I could recommend its publication.

  1. The introduction gave some literature reviews on the mechanical heat treatment and autoclave of municipal waste, without justifying why the author wanted to study the effect of anaerobic digestion temperature, i.e. meo- and thermophilic digestion. There is no logical link between the literature review and the aim of the research.
  2. Much more details needed to be added to the method section. For example, line 85-89, from 58% to 69% is not a decrease in dry matter. The parameters in Table 1 have not been described their determination method and they are ambiguous. COD is not specified, is it soluble COD or total COD, how it is measured?  Total nitrogen normally means the elemental analysis, but from the author, it should be ammonia nitrogen and its measurement was not clear. Total ammonia nitrogen (TAN) needs to be treated by strong alkaline before distillation, not direct distillation. The volume determination of the produced biogas was not mentioned at all. How to measure the table 1 listed dry matter and organic substrate is not described. 
  3. The results were not clearly documented. For instance, in figure 2, there should be 4 trends (or lines), but it is only 2, and I cannot understand what the 2 presented trends stand for. Also, for AD i is normally reported the yields in L CH4 per g VS, not L biogas, because the biogas composition is not the same for variants. For figure 5-6, there are tables inside without any description and caption. Also, the linear regressions do have any sense and in the text, nothing is mentioned about the linear regressions in text at all. The discussion was almost pure speculation, without presents any scientific evidence about the Maillard compounds. 
  4. There are a number of typos and inconsistency in the language. For example, there is a mix of dry-matter and dry matter. It needs to consistent. Also, a few of subscriptions are missing, such as CO2 in line 136. In line 131-132, there are styles of 20 °C/min and 1.0 mL min-1, missing consistency. Throughout the manuscript, litter was written as l, with a few exception of L. in line 238, fosfate should be phosphate and the Phosphorus in table 1 was also wrong. There are many more, more careful proof reading is need to keep consistency.

Author Response

The introduction gave some literature reviews on the mechanical heat treatment and autoclave of municipal waste, without justifying why the author wanted to study the effect of anaerobic digestion temperature, i.e. meo- and thermophilic digestion. There is no logical link between the literature review and the aim of the research.

I am very grateful for this comment. The Introduction section has been completely reorganized and the appropriate explanation has been completed. I hope that the changes made emphasize the novelty of the conducted research.

Lines (Track changes, simple markup): 23-86

Much more details needed to be added to the method section. For example, line 85-89, from 58% to 69% is not a decrease in dry matter. The parameters in Table 1 have not been described their determination method and they are ambiguous. COD is not specified, is it soluble COD or total COD, how it is measured?  Total nitrogen normally means the elemental analysis, but from the author, it should be ammonia nitrogen and its measurement was not clear. Total ammonia nitrogen (TAN) needs to be treated by strong alkaline before distillation, not direct distillation. The volume determination of the produced biogas was not mentioned at all. How to measure the table 1 listed dry matter and organic substrate is not described. 

Table 1 has been improved and the characteristics of substate were supplemented with ammonium results. Total nitrogen given in Table 1 was measured by elemental analysis, which was explained in the “Analytical monitoring of the process” section.

Lines (Track changes, simple markup): 116, 155-158

 The analyses of the organic fraction used in the anaerobic digestion were performed by the Institute of Soil Science and Plant Cultivation (IUNG), State Research Institute in PuÅ‚awy, Poland. In the Analytical monitoring of the process Section, an appropriate explanation was made and the research methodology used by the institute was given.

Lines (Track changes, simple markup): 143-147, 153-158

The results were not clearly documented. For instance, in figure 2, there should be 4 trends (or lines), but it is only 2, and I cannot understand what the 2 presented trends stand for. Also, for AD i is normally reported the yields in L CH4 per g VS, not L biogas, because the biogas composition is not the same for variants. For figure 5-6, there are tables inside without any description and caption. Also, the linear regressions do have any sense and in the text, nothing is mentioned about the linear regressions in text at all. The discussion was almost pure speculation, without presents any scientific evidence about the Maillard compounds. 

Additional calculations of methane yield were made. The results were given, presented in the appropriate graph (Figure 3b) and compared with the literature.

Forster-Carneiro, T.; Pérez, M.; Romero, L. I. Thermophilic anaerobic digestion of source-sorted organic fraction of municipal solid waste. Bioresource technology 2008, 99 (15), 6763-6770.

Zhang, R.; El-Mashad, H. M.; Hartman, K.; Wang, F.; Liu, G.; Choate, C.; Gamble, P. Characterization of food waste as feedstock for anaerobic digestion. Bioresource technology 2007, 98 (4), 929-935.

Lines (Track changes, simple markup): 183-186, 194-200

The trend lines and the tables were removed from the charts.

Lines (Track changes, simple markup): 284-287, 299-301

The paragraph describing the impact of Mailard's compounds on the anaerobic digestion process was improved, which is also in accordance with the comment of another reviewer.

Lines (Track changes, simple markup): 243-256

There are a number of typos and inconsistency in the language. For example, there is a mix of dry-matter and dry matter. It needs to consistent. Also, a few of subscriptions are missing, such as CO2 in line 136. In line 131-132, there are styles of 20 °C/min and 1.0 mL min-1, missing consistency. Throughout the manuscript, litter was written as l, with a few exception of L. in line 238, fosfate should be phosphate and the Phosphorus in table 1 was also wrong. There are many more, more careful proof reading is need to keep consistency.

The manuscript has been checked carefully, and I believe all the errors described have been corrected.

Lines (Track changes, simple markup): whole manuscript

Reviewer 3 Report

Dear Authors,

I revised the manuscript “Mesophilic and Thermophilic Anaerobic Digestion of Organic Fraction Separated during Mechanical Heat Treatment of Municipal Waste.” submitted to the Applied Sciences Journal. The paper is interesting. The article presents the issues of mesophilic and thermophilic methane fermentation of autoclaving municipal waste. The paper is interesting. However, I have some concerns, which need to be addressed.

  1. Introduction

Introduction too short, can be extended.

Describe the others method of thermally treating municipal waste.

Line 30. Please specify the parameters (pressure and temperature) of the autoclave process.

Line 43.  What does the author mean by defibrillates lignocellulosic?

  1. Materials and Methods

2.1 Methods of sampling and characteristics of test material

Line 76. Specify the autoclaving process conditions (temperature and pressure) in MHT plant.

Line 79-84. Why the physicochemical parameters (dry matter, ph, organic dry matter) of these two samples were not given.

Line 89. To measure the AT4 value, 3 repetitions must be made due to the different physical and chemical composition of the organic fraction. Have the measurements been taken for three repetitions. 

2.3 Digestion set-up

Please write down the standard by which biogas yield tests were conducted.

  1. Results and Discussion

It is worth to give the biogas yield in cubic metres per tonne of fresh matter and dry organic matter. Valuable information for the investor.

In Figures 5 and 6, please explain the values attributed to the "k" factors. There is no reference in the text.

Line 197-209. To expand the issue of Maillard's relationship. Replace the compounds formed during Maillard's reaction. Not knowing how the temperature was during the autoclaving process, it is hard to talk about inhibiting the process. Other inhibitors such as furfural, hydroxymethylfurfural (HMF) and phenols formed from the decomposition of sugars and lignin could also be formed during the autoclaving process.

Author Response

Introduction

Introduction too short, can be extended.

Describe the others method of thermally treating municipal waste.

Line 30. Please specify the parameters (pressure and temperature) of the autoclave process.

Line 43.  What does the author mean by defibrillates lignocellulosic?

The Introduction section has been completely reorganized and extended. I hope that the changes made emphasize the novelty of the conducted research. The Introduction section has been supplemented with a drawing showing the technological details of MHT process.

Lines (Track changes, simple markup): 23-86

In particular, “defibrillates lignocellulosic” has been changed to "The process of autoclaving disrupts the lignocellulosic structure in organic waste”.

Lines (Track changes, simple markup): 47-48 

Materials and Methods

2.1 Methods of sampling and characteristics of test material

 Line 76. Specify the autoclaving process conditions (temperature and pressure) in MHT plant.

A detailed description of the technological process was added, including the temperature and pressure range. (Figure 1)

Lines (Track changes, simple markup): 43, 89-96

Line 79-84. Why the physicochemical parameters (dry matter, ph, organic dry matter) of these two samples were not given.

Thank you for this comment. This part of the methodology was described incorrectly. In fact, one sample was taken, which was divided into three parts (1/3 of the research material was intended to be submitted for laboratory analysis). The description of this part of the methodology has been improved.

Lines (Track changes, simple markup): 97-101

Line 89. To measure the AT4 value, 3 repetitions must be made due to the different physical and chemical composition of the organic fraction. Have the measurements been taken for three repetitions. 

 In fact, the measurement was made in triplicate. This paragraph was corrected.

Lines (Track changes, simple markup): 168

2.3 Digestion set-up

Please write down the standard by which biogas yield tests were conducted.

It is worth to give the biogas yield in cubic metres per tonne of fresh matter and dry organic matter. Valuable information for the investor.

Additional calculations of methane yield were made. The results were given, presented in the appropriate graph (Figure 3b) and compared with the literature.

Forster-Carneiro, T.; Pérez, M.; Romero, L. I. Thermophilic anaerobic digestion of source-sorted organic fraction of municipal solid waste. Bioresource technology 2008, 99 (15), 6763-6770.

Zhang, R.; El-Mashad, H. M.; Hartman, K.; Wang, F.; Liu, G.; Choate, C.; Gamble, P. Characterization of food waste as feedstock for anaerobic digestion. Bioresource technology 2007, 98 (4), 929-935. 

Lines (Track changes, simple markup): 183-186, 194-200

Results and Discussion

In Figures 5 and 6, please explain the values attributed to the "k" factors. There is no reference in the text.

Line 197-209. To expand the issue of Maillard's relationship. Replace the compounds formed during Maillard's reaction. Not knowing how the temperature was during the autoclaving process, it is hard to talk about inhibiting the process. Other inhibitors such as furfural, hydroxymethylfurfural (HMF) and phenols formed from the decomposition of sugars and lignin could also be formed during the autoclaving process.

The trend lines and the tables were removed from the charts and the paragraph describing the impact of Mailard's compounds on the anaerobic digestion process was improved, which is also in accordance with the comment of another reviewer.

Lines (Track changes, simple markup): 243-256

Round 2

Reviewer 2 Report

The authors have provided some required details and acceptable response to the comments. 

Author Response

Thank you very much again for the valuable comments.

Reviewer 3 Report

Dear Authors,

I again revised the manuscript “Mesophilic and Thermophilic Anaerobic Digestion of Organic Fraction Separated during Mechanical Heat Treatment of Municipal Waste.” submitted to the Applied Sciences Journal. The work has been improved according to my recommendations. The paper is interesting. I have one more comment.

2.3 Digestion set-up

Please write down the standard by which biogas yield tests were conducted.

  1. Results and Discussion

It is worth to give the biogas yield in cubic metres per tonne of fresh matter and dry organic matter. Valuable information for the investor.

Author Response

2.3 Digestion set-up

Please write down the standard by which biogas yield tests were conducted.

Additional information was given in the 2.2 Test stand section.

Lines (Track changes, simple markup): 121-122

Results and Discussion

It is worth to give the biogas yield in cubic metres per tonne of fresh matter and dry organic matter. Valuable information for the investor.

Additional calculations were made in:

Lines (Track changes, simple markup): 322-324